# Comparative Efficacy and Safety of Tezepelumab and Other Biologics in Patients with Inadequately Controlled Asthma According to Thresholds of Type 2 Inflammatory Biomarkers: A Systematic Review and Network Meta-Analysis

**DOI:** 10.3390/cells11050819

**Published:** 2022-02-26

**Authors:** Koichi Ando, Yosuke Fukuda, Akihiko Tanaka, Hironori Sagara

**Affiliations:** 1Division of Respiratory Medicine and Allergology, Department of Medicine, School of Medicine, Showa University, 1-5-8 Hatanodai, Shinagawa-ku, Tokyo 142-8666, Japan; y.f.0423@med.showa-u.ac.jp (Y.F.); tanakaa@med.showa-u.ac.jp (A.T.); sagarah@med.showa-u.ac.jp (H.S.); 2Division of Internal Medicine, Showa University Dental Hospital Medical Clinic, Senzoku Campus, Showa University, 2-1-1 Kita-senzoku, Ohta-ku, Tokyo 145-8515, Japan

**Keywords:** dupilumab, asthma, tezepelumab, type 2 inflammation, systematic review, network meta-analysis

## Abstract

The anti-thymic stromal lymphopoietin antibody (tezepelumab) has therapeutical potential for inadequately controlled asthma. However, evidence comparing tezepelumab with other biologics is scarce. To address this issue, we performed a network meta-analysis to compare and rank the efficacy of five treatments (tezepelumab, dupilumab, benralizumab, mepolizumab, and placebo) in overall participants and in subgroups stratified by the thresholds of type 2 inflammatory biomarkers, including peripheral blood eosinophil count (PBEC) and fractional exhaled nitric oxide (FeNO). The primary endpoints were annualized exacerbation rate (AER) and any adverse events (AAEs). In the ranking assessment using surface under the cumulative ranking curve (SUCRA) of AER, tezepelumab ranked the highest overall and across subgroups (based on PBEC and FeNO level thresholds). A significant difference was observed between tezepelumab and dupilumab in the patient subgroup with PBEC < 150, and between tezepelumab and benralizumab in overall participants and the patient subgroup with PBEC ≥ 300 and ≥150, respectively. There was no significant difference in the incidence of AAEs in the overall participants between each pair of five treatment arms. These results provide a basis for the development of treatment strategies for asthma and may guide basic, clinical, or translational research.

## 1. Introduction

The widespread use of inhaled corticosteroids (ICS) and combination inhaled therapy, including ICS and bronchodilators containing two or three components, on a global scale have contributed significantly to the management of bronchial asthma [1]. However, even when these standard inhaled therapies are administered at maximum volume, approximately 10% of patients still fail to achieve adequate control [1]. In recent years, biologics targeting pro-inflammatory cytokines or mediators, such as IL-5, IL-4, and Immunoglobulin E (IgE), involved in type 2 inflammation have emerged as therapeutic options for patients whose asthma is uncontrolled by standard inhalation therapy [1,2,3]. However, although these biologics may be effective against asthma caused by type 2 inflammation, they are not always effective for non-type 2 bronchial asthma. Therefore, developing therapeutic strategies for non-type 2 severe bronchial asthma remains a clinically relevant requirement [2,3,4,5].

Thymic stromal lymphopoietin (TSLP) belongs to the IL-2 family and is highly homologous to IL-7. TSLPs are predominately secreted from airway epithelial cells in response to allergens, viruses, bacteria, etc. The TSLP receptor is a heterodimer complex comprising a TSLP receptor (TSLPR) and an IL-7 receptor alpha (IL-7Rα) chain and is highly expressed in dendritic cells. By binding to the TSLPR via Janus kinase (JAK) (mainly JAK1 and JAK2), TSLP activates the signal transducer and activator of transcription (STAT) (mainly STAT5, and STAT1, STAT3, STAT4, and STAT6), phosphatidylinositol-3 kinase (PI3K), c-Jun-N-terminal kinases (JNK), mitogen-activated protein kinase (MAPK), and downstream signaling pathways [6,7,8,9]. This increases the expression of the genes associated with the activation of cells that are involved in a wide range of immune processes and those involved in asthma, inducing various types of airway inflammation and contributing to chronicity and severity (Figure 1) [7,9,10].

Viral infections are known to trigger acute exacerbations of bronchial asthma, and the involvement of TSLP produced by airway epithelial cells in the exacerbation process has been pointed out. The suppression of TSLP is expected to reduce the risk of asthma exacerbations. Therefore, TSLPs have been a longstanding focus of research as targets to treat inadequately controlled asthma [4,6,7,11,12,13,14]. Recently, a phase III trial of an anti-TSLP antibody (tezepelumab) for inadequately controlled bronchial asthma was completed [15]. The results showed that it had a more favorable efficacy than the placebo in groups with high type 2 inflammatory biomarker levels, but also in the group with relatively low biomarker levels [15]. Tezepelumab is expected to be effective, regardless of the level of type 2 inflammatory biomarkers. Accordingly, tezepelumab is expected to be effective for patients with severe, non-type 2 asthma, for whom disease control has traditionally been challenging.

However, there are no previous reports of randomized controlled trials (RCTs) directly comparing tezepelumab with other biologics (including mepolizumab, benralizumab, and dupilumab). It is important to compare the efficacy and safety of tezepelumab with those of other biologic agents according to levels of type 2 inflammatory biomarkers to improve treatment strategies for patients with bronchial asthma, including those with severe non-type 2 asthma. Such comparative analyses will provide important information for clinical decision-making, especially in cases of severe non-type 2 asthma, and for reforming and revising bronchial asthma treatment strategies. Although RCTs are the most appropriate means of comparing the safety and efficacy of treatments, large-scale RCTs require a great deal of time and effort as well as considerable implementation costs.

In this study, we adopted a network meta-analysis (NMA) approach to compare the safety and efficacy of various therapeutic agents [16,17,18]. By performing NMA, it is possible to make comparisons for comparison groups (for which there are no previous direct RCTs) and rank each treatment group [16,17,18,19,20,21,22,23,24]. In this systematic review and NMA (registration: UMIN-CTR number UMIN000044672), we used Bayesian NMA statistical methods [22,25] to compare and rank the efficacy and safety of five treatments (i.e., tezepelumab, dupilumab, benralizumab, mepolizumab, and placebo) in patients with inadequately controlled bronchial asthma. Moreover, the efficacy was evaluated and ranked according to threshold levels of type 2 inflammatory biomarkers, i.e., the peripheral blood eosinophil counts (PBECs) and fractional exhaled nitric oxide (FeNO) levels. We focused on the comparison of the efficacy and safety of tezepelumab with those of dupilumab for inadequately controlled bronchial asthma with respect to type 2 inflammation biomarker thresholds. This is because among biologic agents, dupilumab seemed to be an appropriate comparator to assess the efficacy and safety of tezepelumab in non-type 2 asthma, as the involvement of type 2 inflammation in asthma pathogenesis has been suggested to correlate well with the efficacy of dupilumab. Moreover, in contrast to benralizumab and mepolizumab, whose pharmacological effects are limited to eosinophilic inflammation involving IL-5, dupilumab comprehensively suppresses Th2-dominated type 2 inflammation by inhibiting signaling involving IL-4 and IL-13 [3,4,5,12,13].

The aim of the study was to compare the efficacy and safety of tezepelumab and other biologics in patients with inadequately controlled asthma according to the thresholds of type 2 inflammatory biomarkers.

## 2. Materials and Methods

### 2.1. Systematic Review

We conducted a comprehensive and exhaustive literature search to identify reports published from 1946 to 26 July 2021, in four databases (PubMed [26], Cochrane Library [27], EMBASE [28], and SCOPUS [29]). For the search strategy, various keywords were used, including tezepelumab, dupilumab, benralizumab, mepolizumab, asthma, and their medical subject heading (MeSH) terms. The search formula used in PubMed is shown in Appendix B, and the same strategy was used for all four databases. To ensure a comprehensive survey, the reference lists of the retrieved studies were checked for relevant studies that met the inclusion criteria. Furthermore, a manual search was conducted to identify all associated research and minimize publication bias. If the reported outcome data was deemed to be insufficient, the corresponding authors were contacted by e-mail, if necessary. The primary aim of this systematic review was to identify all associated published RCTs. to compare and rank the safety and efficacy of tezepelumab, dupilumab, benralizumab, and mepolizumab and the placebo in patients with inadequately controlled asthma. This study was conducted in accordance with the Preferred Reporting Items for Systematic Reviews and Meta-Analyses (PRISMA) guidelines [30] and the PRISMA Extension Statement for Reporting of Systematic Reviews Incorporating Network Meta-analyses (PRISMA-NMA) [31,32]. Two researchers (K.A. and Y.F.) independently conducted the literature search. To address clinical or methodological heterogeneity among studies and to secure the validity of indirect treatment comparison, the PICOS (patient, intervention, comparison, outcome, and study design) approach was adopted for the retrieved studies.

### 2.2. Quality Evaluation

The quality of the RCTs in the NMA was assessed using the Cochrane risk of bias tool version 2 (RoB 2) [33]. The following parameters were assessed as either low risk, some concerns, or high risk: (1) bias arising from the randomization process, (2) bias due to deviations from intended interventions, (3) bias due to missing outcome data, (4) bias in measurement of outcomes, and (5) bias in the selection of the reported results. Evaluations were conducted independently by two investigators (K.A. and Y.F.) and in case of conflicts we consulted with a third researcher (A.T.).

### 2.3. Inclusion and Exclusion Criteria (Predefined PICOS)

#### 2.3.1. Patients

The inclusion criteria were as follows: (1) a minimum age of 12 years, (2) inadequately controlled asthma requiring moderate-to-high doses of ICS, and (3) at least one episode of exacerbation in the previous year.

#### 2.3.2. Interventions and Comparisons

The NMA included data for tezepelumab (210 mg, subcutaneously, once every 4 weeks), dupilumab (300 mg, subcutaneously, once every 2 weeks), benralizumab (30 mg, subcutaneously, once every 8 weeks after the first three doses administered once every 4 weeks), and mepolizumab (100 mg, subcutaneously, once every 4 weeks). These doses were the approved dosage, or the dosage employed in the phase III study. Studies that included these treatments were included in the analysis. The placebo was also considered a common comparator for each treatment.

#### 2.3.3. Outcomes

The primary efficacy endpoint was the annualized exacerbation rate (AER), for which the corresponding rate ratio (RR) and 95% credible interval (CrI) were calculated. The secondary efficacy endpoints were a change in pre-bronchodilator forced expiratory volume in one second (pre-BD FEV_1.0_), change in asthma control questionnaire (ACQ) score, and change in Asthma Quality of Life Questionnaire (AQLQ) score. The corresponding mean difference (MD) (change in pre-BD FEV_1.0_) or standardized mean difference (SMD) and their 95% CrIs were calculated. The primary safety endpoint was the frequency of any adverse events (AAEs), for which the corresponding odds ratios (ORs) and 95% CrIs were calculated. To rank the safety and efficacy of each treatment, we calculated the value of the surface under the cumulative ranking curve (SUCRA) for each outcome [34]. The SUCRA indicates the area of under the curve of the cumulative ranking of probabilities for each treatment group, expressed in the range of 0–100%. A higher SUCRA value indicates that a treatment yields a more favorable endpoint. The SUCRA is an index that can be used as a reference to evaluate the relative position of each treatment and accounts for inconsistencies between studies [34]. The SUCRA values alone cannot verify the significance of the difference in efficacy of the drugs. Therefore, it is not possible to make a final conclusion about these differences [34]. For inclusion, at least one predefined efficacy or safety endpoint had to be assessed in the study. These defined efficacy and safety endpoints were analyzed only when relevant data were available from the trials included in the present study. Two authors (K.A. and Y.F.) independently extracted the relevant data and, when discrepancies occurred, consulted with the third author (A.T.) to address the discrepancies as necessary.

#### 2.3.4. Study Design

Parallel group phase III or IIIb RCTs were considered for inclusion.

### 2.4. Statistical Analysis Methods

A Bayesian NMA was conducted following a robustly established methodology outlined by The National Institute for Health and Care Excellence (NICE) [20,21], employing the standard Bayesian model described by Dias et al. [35,36,37], which accounted for between-study inconsistency and heterogeneity. Applying a non-informative prior distribution, the posterior distribution of the effect size was estimated by using the Gibbs sampling technique based on the Markov chain Monte Carlo method [25]. The number of iterations was set to 50,000, and the first 10,000 iterations were used as a burn-in sample to eliminate the effect of the initial values. The effect size was expressed as RR, MD, SMD, OR, and 95% CrI, and the difference in effect size between the treatment groups for each endpoint was considered significant if the 95% CrI did not include 1 (RR or OR) or 0 (MD or SMD). The SUCRA values ranged from 0% to 100%, with larger values indicating better treatment outcomes [34]. The Brooks–Gelman–Rubin (BGR) diagnostic method [38,39] was also used to evaluate convergence for all comparisons. Both visual and BGR diagnostics were used to check for convergence. OpenBUGS 1.4.0 (MRC Biostatistics Unit, Cambridge Public Health Research Institute, Cambridge, UK) was used for the analysis, and GraphPad Prism (ver. 9.) (GraphPad Software, Waco, CA, USA) was used to visualize the results.

### 2.5. Sensitivity Analysis

In cases of statistically significant heterogeneity between studies, the relevant studies were excluded. In addition, sensitivity analyses [40,41] were performed to determine whether the inclusion/exclusion of studies deemed to have conceptual differences affected the final conclusions.

### 2.6. Assessment of Inter-Study Heterogeneity

To consider inter-study heterogeneity, we evaluated the *I*^2^ statistic (%) [42]. An *I*^2^ statistic greater than 50% indicated that high heterogeneity existed between the studies [42].

### 2.7. Ethical Aspects

Institutional review board approval and patient consent were waived owing to the nature of this systematic review.

## 3. Results

### 3.1. Systematic Review

In our systematic literature review, we identified 1507 studies (276 from PubMed, 196 from EMBASE, 661 from CENTRAL, and 374 from SCOPUS), and 840 articles were retained after the removal of duplicates. After adopting the PICOS approach, eight studies were selected for the NMA. Two studies compared mepolizumab with placebo (MENSA [43] and MUSCA [44]), four studies compared benralizumab with placebo (SIROCCO [45], CALIMA [46], SOLANA [47], and ANDHI [48]), and one study each compared dupilumab with placebo and tezepelumab with placebo (LIVERTY ASTHMA QUEST [49] and NAVIGATOR [15], respectively). The process adopted for the study selection is represented in Figure 2, the list of studies and the main inclusion criteria for each included in the present analysis are shown in Appendix A, and the main characteristics of the included studies are presented in Appendix A. The studies included in the analysis for each outcome are summarized in Appendix A. Data for all 5524 patients were analyzed (corresponding to eight studies, placebo: 2511, mepolizumab: 468; benralizumab, 1384; dupilumab, 633; and tezepelumab, 528). A network map of this analysis is presented in Figure 3.

### 3.2. Primary Efficacy Endpoint: AER

There were no significant differences in AERs between patients treated with tezepelumab and those treated with dupilumab (RR, 0.815; 95% CrI, 0.609–1.092) in overall participants. Similar results were obtained for the patient groups with PBEC of ≥300 cells/mm^3^, <300 cells/mm^3^, and ≥150 cells/mm^3^. However, in the subgroup with a PBEC of <150 cells/mm^3^, the AER was significantly better in the tezepelumab-treated group than in the dupilumab-treated group (RR, 0.531; 95% CrI, 0.302–0.939) (Figure 4). In the analysis stratified by FeNO thresholds, there were no significant differences in the AERs between tezepelumab and dupilumab treatments in groups with FeNO ≥ 50, <50, ≥25, and <25 ppb (Figure 4).

For the overall participants and subgroups based on the PBEC, comparisons of AERs between each pair of treatment groups (tezepelumab, dupilumab, benralizumab, mepolizumab, and placebo), including the comparison between the tezepelumab and dupilumab groups shown in Figure 4, are summarized in Appendix A and the SUCRA results are shown in Appendix A.

In the comparison between tezepelumab and benralizumab, the AER was significantly better in the tezepelumab-treated group than in the benralizumab-treated group in overall participants and in the subgroups with PBEC ≥ 300 and ≥150. With respect to the comparison of tezepelumab with mepolizumab, no significant differences in efficacy in the AER were observed in the overall participants and in all subgroups of PBEC thresholds.

In pairwise comparisons among the three existing biologics other than tezepelumab, benralizumab had a significantly higher AER than mepolizumab in the overall participants. Furthermore, dupilumab and mepolizumab had significantly better annual asthma exacerbation rates than benralizumab in the subgroups with PBEC ≥ 300 and ≥150, respectively.

Based on the SUCRA values, tezepelumab ranked the highest followed by mepolizumab, not only in the overall participants, but also in each subgroup (i.e., with PBEC of ≥300, <300, ≥150, and <150).

A comparison of AERs by FeNO thresholds was only possible between three treatment groups (tezepelumab, dupilumab, and placebo). The results of these comparisons, including the comparison between the tezepelumab and dupilumab-treated patient groups shown in Figure 4, are summarized in Appendix A and SUCRA results are presented in Appendix A. Based on SUCRA values, tezepelumab ranked the highest in terms of AER in each subgroup (i.e., with FeNO of ≥50, <50, ≥25, and <25).

### 3.3. Secondary Efficacy Endpoint: Change in Pre-BD FEV_1.0_

There were no significant differences in the change in pre-BD FEV_1.0_ between tezepelumab and dupilumab (MD, 0.000; 95% CrI, −0.071 to 0.071) in the overall participants. Similar results were obtained for subgroups with PBEC of ≥300, <300, ≥150, and <150 cells/mm^3^ (Figure 5).

For the overall participants and subgroups by PBEC thresholds, comparisons in terms of the change in pre-bronchodilator FEV_1.0_ between each pair of treatment groups (tezepelumab, dupilumab, benralizumab, mepolizumab, and placebo), including the comparison between the tezepelumab and dupilumab groups shown in Figure 5, are summarized in Appendix A and the SUCRA results are shown in Appendix A.

There was no significant difference in change in pre-BD FEV_1.0_ between tezepelumab and benralizumab in the overall participants and in the subgroups with PBEC < 300, ≥150, and <150. However, in the subgroup of PBEC ≥300, the change in pre-BD FEV_1.0_ was significantly better in the tezepelumab group than in the benralizumab group.

Owing to insufficient reported data, comparisons between mepolizumab and other biologics were only possible in overall participants. There was no significant difference in the change in pre-BD FEV_1.0_ between tezepelumab and mepolizumab in the overall participants.

There was no significant difference in the change in pre-BD FEV_1.0_ between dupilumab and benralizumab in the overall participants and in the subgroups with PBEC < 300, ≥150, and <150. However, in the subgroup of PBEC ≥ 300, the change in pre-BD FEV_1.0_ was significantly better in the dupilumab group than in the benralizumab group. There was no significant difference in pre-BD FEV_1.0_ between dupilumab and mepolizumab, and between benralizumab and mepolizumab.

The SUCRA values for pre-BD FEV1.0 were the highest in the tezepelumab group in the overall group and in the subgroups with PBEC < 300 and ≥150, and the highest in the dupilumab group in the subgroups with PBEC ≥ 300 and <150, respectively.

In regard to FeNO thresholds, data were insufficient for comparisons between the five treatment groups (including comparisons between tezepelumab and dupilumab).

### 3.4. Secondary Efficacy Endpoint: Change in the AQLQ Score

There were no significant differences in the change in AQLQ score between patients treated with tezepelumab and those treated with dupilumab (SMD, 0.120; 95% CrI, −0.062 to 0.300) in overall participants. Similar results were obtained for the patient group with PBEC ≥ 300 cells/mm^3^ (Figure 6).

Owing to insufficient data, pairwise comparisons for all participants and subgroups with PBEC of ≥300 cells/mm^3^ were only possible for four treatment groups: tezepelumab, dupilumab, benralizumab, and placebo (mepolizumab could not be included in the analysis). Pairwise comparisons for the subgroup with PBEC of ≥150 cells/mm^3^ were only possible for three treatment groups: tezepelumab, benralizumab, and placebo. Appendix A shows these results, including the comparison between tezepelumab and dupilumab shown in Figure 6. Ranking results based on SUCRA scores in terms of the change in the AQLQ score are shown in Appendix A.

There was no significant difference between the tezepelumab and benralizumab groups in the overall group or in the group with PBEC ≥ 300, but in the subgroup with PBEC ≥ 150, the change in AQLQ score was significantly better in the tezepelumab group than in the benralizumab group.

There was no significant difference in the change in AQLQ score between the dupilumab and benralizumab groups in the overall group or in the group with PBEC ≥ 300.

In the groups with PBEC of <300 and <150, comparisons of the change in AQLQ score were not possible. Likewise, we could not compare treatment groups with respect to the change in AQLQ score in the groups with FeNO ≥ 50, <50, ≥25, and ≥25 ppb.

In the SUCRA-based ranking assessment of the four treatments (tezepelumab, dupilumab, benralizumab, and placebo), tezepelumab ranked the highest in the overall population and in each subgroup with PBEC ≥ 300 and ≥150, although dupilumab could not be included in the comparison of the subgroup with PBEC ≥ 150.

### 3.5. Secondary Efficacy Endpoint: Change in the ACQ Score

There were no significant differences in the change in ACQ score between patients treated with tezepelumab and those treated with dupilumab (SMD, −0.160; 95% CrI, −0.337 to 0.018) in the overall participants. Owing to the reporting of insufficient data, it was not possible to conduct a subgroup analysis based on PBEC or FeNO levels for the comparison between tezepelumab and dupilumab.

Comparisons among groups with PBEC of ≥300 cells/mm^3^, <300 cells/mm^3^, and ≥150 cells/mm^3^ could only be performed for each pair of the three treatment groups: tezepelumab, benralizumab, and placebo. Appendix A summarizes the results. The ranking analysis in terms of the change in the ACQ score is shown in Appendix A.

There was no significant difference between the tezepelumab and benralizumab groups in the overall participants group, in the subgroup with PBEC ≥ 300, or in the subgroup with PBEC < 300. However, in the subgroup with PBEC ≥ 150, the change in the ACQ score was significantly better in the tezepelumab group than in the benralizumab group.

In the overall participants group, dupilumab was less effective in terms of change in the ACQ score than mepolizumab, although there was no significant difference between dupilumab and benralizumab.

In the groups with PBEC < 150 cells/mm^3^, comparisons of the change in the ACQ score between treatment groups were not possible. Similarly, we could not compare the change in ACQ score in the subgroups with FeNO ≥ 50, <50, ≥25, and ≥25 ppb.

In the SUCRA-based ranking assessment of the change in ACQ scores, mepolizumab was ranked the highest in the overall participant group, followed by tezepelumab, benralizumab, dupilumab, and placebo. In addition, tezepelumab was ranked the highest in the patient subgroup with PBEC ≥ 300, <300, and ≥150, respectively.

### 3.6. Primary Safety Endpoint: Incidence of AAEs

There were no significant differences in the incidence of AAEs between patients treated with tezepelumab and those treated with dupilumab (OR, 0.964; 95% CrI, 0.604–1.547) in the overall participants. The results of pairwise comparisons between the five treatment groups in all participants, including a comparison of tezepelumab and dupilumab, are summarized in Appendix A.

Regarding the incidence of AAEs, the results showed no significant difference in the incidence of AAEs between each pair of any treatment groups, namely the four treatment groups (tezepelumab, dupilumab, benralizumab, and mepolizumab groups).

Owing to insufficient data reported, it was not possible to compare the incidence of AAEs in subgroups based on PBEC of FeNO levels between the five treatment pairs, including between tezepelumab and dupilumab. The SUCRA values for the incidence of AAEs were the most favorable for mepolizumab, followed by tezepelumab, dupilumab, benralizumab, and placebo.

### 3.7. Sensitivity Analysis

One study (SOLANA) [47] had a shorter study period (12 weeks) than the other included studies. The average study duration for the other seven studies was 41.1 weeks. SOLANA [47] was not included in the analysis of AER, the primary efficacy endpoint, but was included in the analysis of the change in FEV_1.0_. Accordingly, we performed a sensitivity analysis. By excluding SOLANA [47] in the analysis of the change in FEV_1.0_ in overall participants, the results were unchanged for all pairwise comparisons between the five drug groups (Appendix A). Therefore, the inclusion of SOLANA [47] did not affect our final conclusions.

### 3.8. Quality Evaluation

We assessed the quality of the included studies based on the Cochrane-recommended RoB 2 [33]. In one study [47], we identified “some concerns” in the area of bias arising from the randomization process due to an inadequate description of randomization. However, there were no studies judged to be high risk (Appendix A).

### 3.9. Between-Study Heterogeneity

Two studies (MENSA [43] and MUSCA [44]) compared mepolizumab with a placebo. Therefore, between-study heterogeneity for the primary endpoint, AER, was statistically assessed. The *I*^2^ value was 0.0% (*p =* 0.602), indicating no statistically significant between-study heterogeneity (Appendix A).

Similarly, SIROCCO [45], CALAIMA [46], and ANDHI [48] compared benralizumab and placebo. Here, the *I*^2^ value was 25.7% (*p* = 0.260), indicating there was no significant between-study heterogeneity (Appendix A).

## 4. Discussion

Here, we conducted an NMA of five treatment groups (tezepelumab, dupilumab benralizumab, mepolizumab, and placebo groups) to compare the efficacy and safety of tezepelumab and other biologics in patients with inadequately controlled asthma based on their thresholds for PBEC or FeNO levels. There were no significant differences in AERs between tezepelumab and dupilumab in all participants, subgroups with PBEC of ≥300 cells/mm^3^, <300 cells/mm^3^, and ≥150 cells/mm^3^, and subgroups with FeNO of ≥50 ppb, <50 ppb, ≥25 ppb, and <25 ppb. However, in the subgroup with PBEC of <150 cells/mm^3^, the AER was significantly better in the tezepelumab group than in the dupilumab group. In terms of efficacy based on the AER, although there was no significant difference between tezepelumab and mepolizumab for the overall participants group and for all subgroups analyzed according to PBEC thresholds, in the overall population and in subgroups with PBEC ≥ 300 and ≥150, the tezepelumab group showed a significantly better efficacy profile than the benralizumab group.

In the analysis of the five treatment arms, tezepelumab ranked the highest in terms of SUCRA of efficacy based on the AER, both overall and across subgroups (based on PBEC and FeNO level thresholds). Regarding the safety endpoint, there was no significant difference in the incidence of AAEs between each pair of the four treatment arms in overall participants.

Several biologics are widely used in clinical practice and are highly effective in patients who do not respond adequately to conventional inhalation therapy or oral steroid maintenance therapy [1,3,13,50]. However, the efficacy of available biologics depends on type 2 inflammation and, thus, their efficacy in non-type 2 asthma is limited [3,4,5,50,51]. Therefore, detailed treatment strategies for refractory non-type 2 asthma are not yet fully developed. In this milieu, in a recent phase III trial, tezepelumab was shown to have a better efficacy profile the placebo, irrespective of the PBEC [15]. Therefore, tezepelumab is expected to be effective in patients with non-type 2 asthma. However, to evaluate the efficacy of tezepelumab in patients with non-type 2 bronchial asthma in more detail, it is necessary to compare the efficacy of tezepelumab with conventional biologics, including dupilumab, benralizumab, and mepolizumab, in patients with non-type 2 asthma. Previous various NMAs have indirectly compared the efficacy of several biologics for inadequately controlled asthma [52,53,54,55,56,57,58]. However, tezepelumab has not been compared with other biologics according to type 2 inflammatory biomarker levels. The validation of our results would provide valuable information for the development of treatment strategies for refractory non-Type 2 asthma. We focused on the comparison of tezepelumab and dupilumab among each pairwise comparison. This was because dupilumab comprehensively suppresses Th2-dominated type 2 inflammation by inhibiting signaling involving IL-4 and IL-13. Moreover, the degrees of involvement of type 2 inflammation in asthma pathogenesis have been suggested to correlate well with the efficacy of dupilumab [3,4,5,12,13,54]. To the best of our knowledge, for the first time, we independently compared the efficacy of tezepelumab with other biologics using the PBEC and FeNO thresholds. To our knowledge, the current study provides the first evidence for tezepelumab exhibiting better efficacy than that exhibited by dupilumab based on the AER in a subgroup of patients with low PBEC (<150 cells/mm^3^). Furthermore, while tezepelumab and mepolizumab did not differ significantly in the efficacy based on the AER in the overall participant population and in all subgroups analyzed by PBEC threshold, the tezepelumab group had a significantly better efficacy profile than the benralizumab group in the overall participant population and in the PBEC ≥ 300 and ≥150 subgroups.

More notably, our results revealed tezepelumab ranks the highest (as determined by SUCRA values) for efficacy based on the AER among the five treatment groups (i.e., tezepelumab, dupilumab, benralizumab, mepolizumab, and placebo), not only in the overall participant population, but also in subgroups stratified by the PBEC and FeNO thresholds. Our results suggest that tezepelumab is a promising treatment option for both non-type 2 and type 2 bronchial asthma, with high efficacy independent of the type 2 inflammatory status.

These results may be explained from a molecular biology perspective. TSLPs promote the differentiation of Th0 cells into Th2 cells via dendritic cells expressing TSLPRs [6,9,10,59]. Th2 cells, along with cytokines (such as interleukin [IL]-4, IL-13, and IL-5), play a central role in type 2 inflammation [6,8,9,10,12,14,59,60]. TSLPs also promote the differentiation of Th0 cells into Th17 cells via IL-1β, TGF-β, and IL-6. Th17 cells act on airway epithelial cells, induce neutrophilic airway inflammation, and play a central role in the pathogenesis of non-type 2 bronchial asthma [3,7,10,11,13,61]. TSLP is deeply involved in the pathogenesis of type 2 as well as non-type 2 bronchial asthma. Based on these functions, TSLP is a candidate therapeutic target not only in type 2 asthma, but also in non-type 2 asthma. Dupilumab targets IL-4Rα and inhibits signaling involving IL-4 or IL-13, thereby suppressing type 2 inflammation [62,63]. Benralizumab and mepolizumab suppress eosinophilic airway inflammation by inhibiting the effects of IL-5 and have excellent suppressive effects on type 2 inflammation [64,65]. However, the inhibitory effects of dupilumab, benralizumab, and mepolizumab on non-type 2 inflammation are very limited [5]. These differences in the effects of tezepelumab and conventional biologics (dupilumab, benralizumab, and mepolizumab) on non-type 2 inflammation may explain our findings (Figure 7) [2,3,4,66,67].

In the ranking assessment, tezepelumab presented the highest SUCRA in terms of efficacy for both change in pre-BD FEV_1.0_ and change in ACQ score in the overall participants. Based on the ACQ score, tezepelumab ranked the best in not only the overall participants, but also all PBEC subgroups. It is difficult to assess the significance of drug efficacy based on the SUCRA values alone. However, at least, these results suggest that tezepelumab is a promising treatment option for controlling exacerbation, improving pulmonary function, and reducing asthma symptoms.

Although tezepelumab (SUCRA, 74.1) ranked highest in the rank-based assessment of efficacy for the change in pre-BD FEV_1.0_, dupilumab (SUCRA, 74.0) also performed relatively well. This may explain why dupilumab targets IL-13, which plays a central role in bronchial smooth muscle contraction, fibrosis, and basement membrane thickening, leading to decreased lung function [68,69,70].

The subgroup analysis revealed that in the low PBEC group, AER was significantly more favorable in the tezepelumab group than in the dupilumab group, whereas in the low FeNO group, there was no significant difference in AER between tezepelumab and dupilumab. The patients in the analysis were regular users of inhaled steroids in the baseline phase of the clinical trial. Because FeNO levels show greater fluctuations in response to these inhaled medications than those of PBEC [71,72], it is possible that the low FeNO group included a relatively large number of patients not only with non-type 2 asthma, but also with type 2 asthma, which may have influenced the results.

Our study findings do not necessarily imply that if tezepelumab becomes available to treat bronchial asthma, it should be recommended as a first-line treatment for both inadequately controlled type 2 and non-type 2 patients with asthma. Developing a detailed therapeutic strategy to optimize the use of biologics in patients with inadequately controlled bronchial asthma is warranted.

As this was an NMA that included direct and indirect comparisons, it is difficult to draw definitive conclusions based on its results. However, we observed tezepelumab had a favorable efficacy profile, especially in patients with non-type 2 asthma, and was well-tolerated. Our results provide useful information for clinical practitioners involved in developing treatment strategies for non-type 2 asthma and to guide future basic, clinical, and translational research.

Several limitations of this NMA should be recognized. First, this was a systematic review, and the NMA compared the efficacy and safety of tezepelumab and dupilumab using type 2 inflammatory biomarker thresholds. However, the analysis based on FeNO thresholds was valid only for AERs and not for other outcomes. Moreover, among the biomarkers of type 2 inflammation, only PBEC and FeNO levels were employed in this analysis. Threshold-specific analyses of other biomarkers of type 2 inflammation (such as IgE and periostin) were not performed in this study. To evaluate the efficacy of tezepelumab in non-type 2 asthma, it is necessary to evaluate multiple outcomes based on multiple biomarkers, not just AERs. Second, the current analysis involved studies that included a group of patients with a history of previous exacerbations. Seven studies [15,43,44,45,46,47,48] used a history of at least two exacerbations as a criterion for patient inclusion and one [49] used a history of at least one exacerbation. We cannot exclude the possibility that these differences in inclusion criteria between studies had a non-negligible impact on our results. Third, in this study, the SUCRA was used to rank the drugs. However, it should be noted that it is difficult to refer to the significance of drug efficacy based on the SUCRA values alone, and the ranking of SUCRA values is only an auxiliary reference finding. The comparison of efficacy and safety among drugs should be evaluated based on statistical significance. Finally, regarding the study duration, there was variation among the included studies. Although a sensitivity analysis showed that the inclusion/exclusion of a study that differed substantially with respect to study duration [47] did not significantly affect the final conclusions, we cannot exclude the effect of differences in the study duration.

## 5. Conclusions

We compared the efficacy and safety of tezepelumab and dupilumab in patients with inadequately controlled asthma according to thresholds of biomarkers of type 2 inflammation (PBEC or FeNO) by an NMA. There was no significant difference in AERs between tezepelumab and dupilumab in all participants, subgroups with PBEC of ≥300 cells/mm^3^, <300 cells/mm^3^, and ≥150 cells/mm^3^, and subgroups with FeNO of ≥50 ppb, <50 ppb, ≥25 ppb, and <25 ppb. However, in the subgroup with peripheral blood eosinophils <150 cells/mm^3^, the AER was significantly better in the tezepelumab group than in the dupilumab group. There was no significant difference in efficacy based on the AER between tezepelumab and mepolizumab in the overall participant group and in all subgroups analyzed according to the PBEC threshold, whereas the result of the comparison between tezepelumab and benralizumab showed that the tezepelumab group had a significantly better efficacy profile than the benralizumab group in the overall population and in subgroups with PBEC ≥ 300 and ≥150. The SUCRA analysis of the same five treatment groups showed that among the overall participants and in overall subgroups, the SUCRA value for tezepelumab ranked the highest in terms of efficacy based on the AER. In terms of safety, there was no significant difference in the incidence of AAE between each pair of the five treatment arms. These results suggest that tezepelumab may be an effective treatment option for patients with inadequately controlled bronchial asthma, regardless of the status of type 2 inflammation involvement in their pathogenesis. The findings further provide important information for developing treatment strategies for inadequately controlled bronchial asthma, especially non-type 2 asthma. Furthermore, the results of this study can serve as a foundation for future basic, clinical, and translational research. Considering that this was an NMA comprising direct and indirect comparisons, further clinical trials, such as large head-to-head trials with direct comparisons, are required to confirm the results.

## Figures and Tables

**Figure 1 cells-11-00819-f001:**
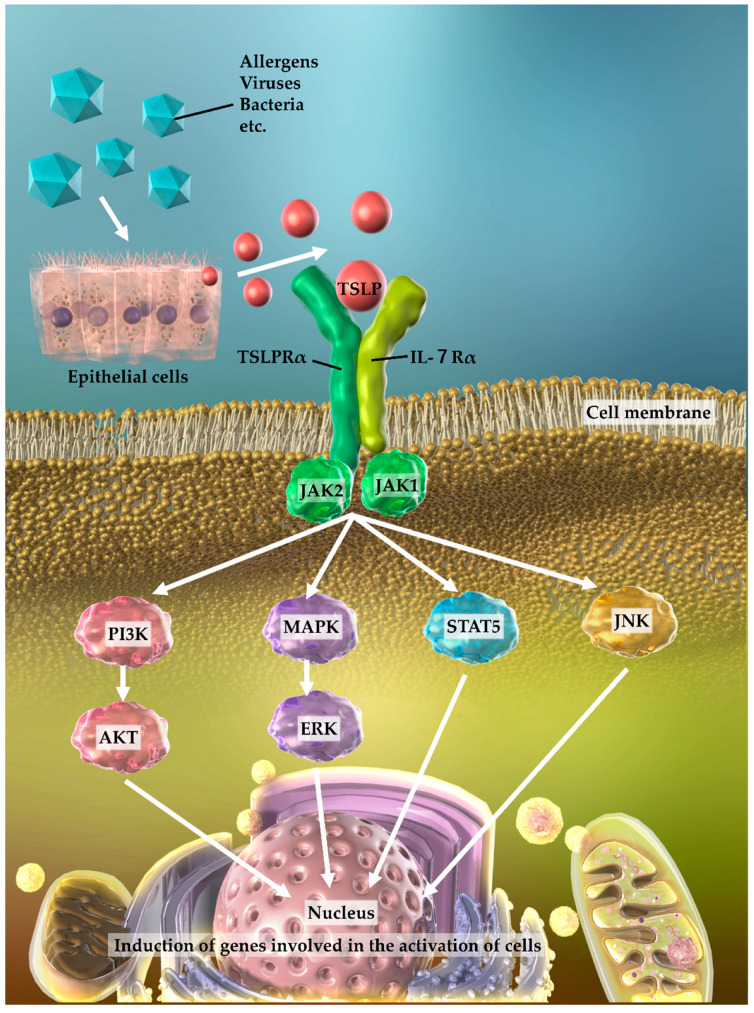
Thymic stromal lymphopoietin (TSLP) production, receptor structure, and intracellular signaling. TSLPs are produced by airway epithelial cells upon exposure to environmental agents, such as allergens and viruses. TSLPs bind to TSLP receptors (heterodimers comprising a TSLP receptor [TSLPR] and an IL-7R alpha [IL-7Rα] chain) expressed on dendritic cells and activate signal transducer and activator of transcription (STAT), mitogen-activated protein kinase (MAPK), phosphatidylinositol-3 kinase (PI3K), c-Jun-N-terminal kinases (JNK), and downstream signaling factors. They are involved in the expression of genes that induce the activation of cells, resulting in various types of airway inflammation and contributing to their chronicity and severity; JAK, Janus kinase; AKT, Ak strain transforming; ERK, extracellular signal-regulated kinase.

**Figure 2 cells-11-00819-f002:**
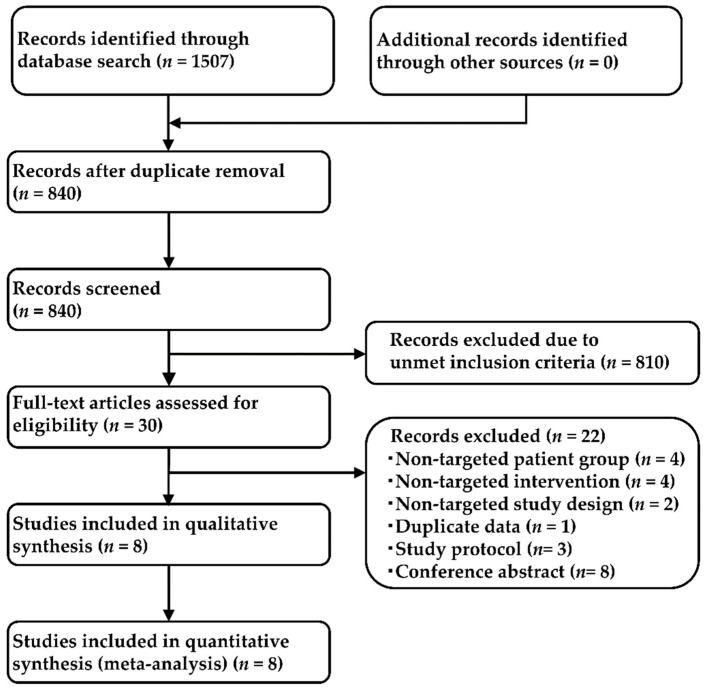
Flow chart depicting the process used for study selection. A total of 1507 studies (276 from PubMed, 196 from EMBASE, 661 from CENTRAL, and 374 from SCOPUS) were detected using a systematic literature review. After eliminating duplicates, 840 references remained; eight studies were finally selected for the NMA after adopting the PICOS approach.

**Figure 3 cells-11-00819-f003:**
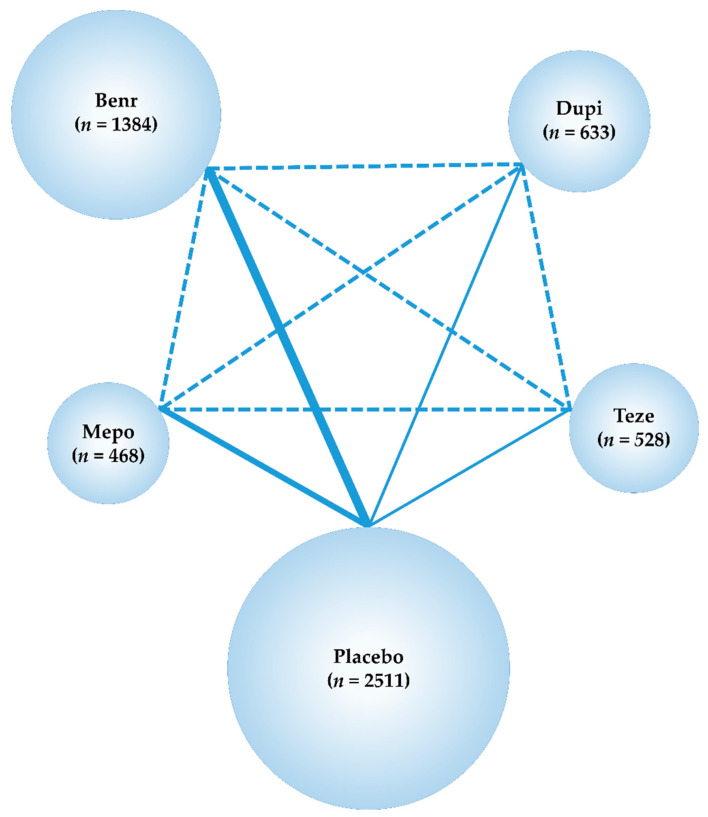
Network map of the five treatment arms tezepelumab, dupilumab, benralizumab, mepolizumab, and placebo. The RCTs included in the present network meta-analysis are indicated by solid lines, and the width of the solid line represents the number of included studies. Dashed lines indicate that there were no previous head-to-head RCTs and that indirect treatment comparisons can be attempted. *n* is the number of participants in each treatment arm. Teze, Tezepelumab; Dupi, dupilumab; Benr, benralizumab; Mepo, mepolizumab; RCT, randomized controlled trial.

**Figure 4 cells-11-00819-f004:**
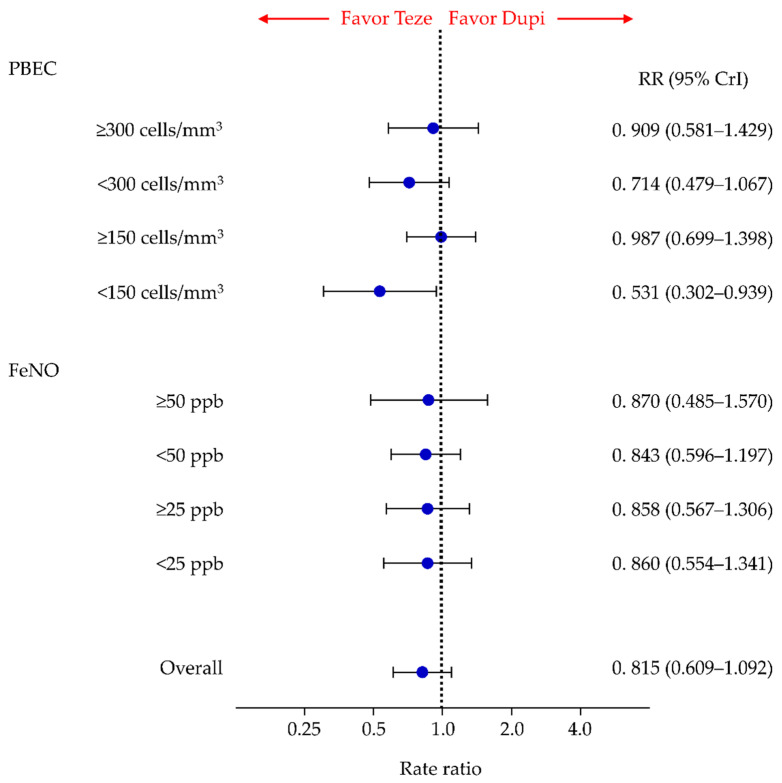
Comparative efficacy of tezepelumab and dupilumab in terms of AERs in patients with inadequately controlled asthma for all participants and subgroups stratified by threshold levels of PBEC and FeNO. Comparisons are expressed as tezepelumab versus dupilumab. Data are expressed as rate ratios (RRs) and 95% credible intervals (CrIs). AER, annualized exacerbation rate; Teze, tezepelumab; Dupi, dupilumab; PBEC, peripheral blood eosinophil counts; FeNO, fractional exhaled nitric oxide.

**Figure 5 cells-11-00819-f005:**
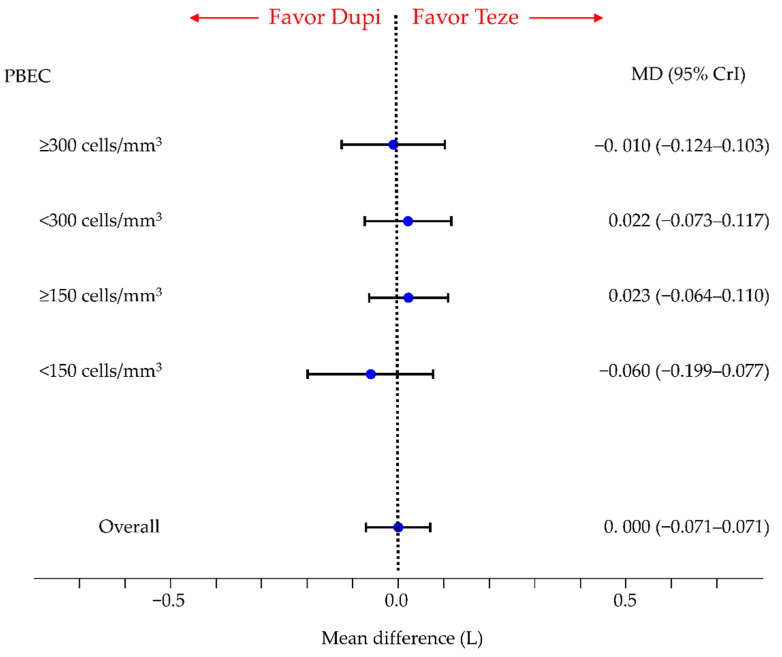
Comparative efficacy of tezepelumab and dupilumab in terms of the change in pre-BD FEV_1.0_ in patients with inadequately controlled asthma in overall participants and subgroups based on PBEC thresholds. Comparisons are expressed as tezepelumab versus dupilumab. Data are expressed as mean differences (MDs) and 95% credible intervals (CrIs). pre-BD FEV_1.0_, pre-bronchodilator forced expiratory volume in one second; Teze, tezepelumab; Dupi, dupilumab; PBEC, peripheral blood eosinophil counts.

**Figure 6 cells-11-00819-f006:**
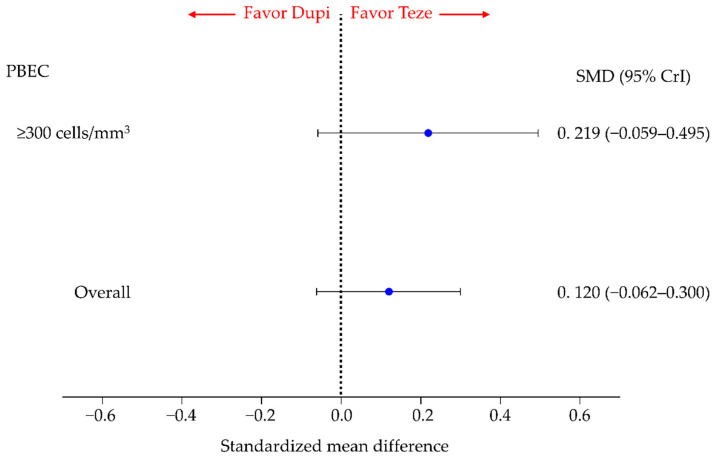
Comparative efficacy of tezepelumab and dupilumab in terms of the change in the AQLQ score in patients with inadequately controlled asthma in overall participants and subgroups stratified by PBEC thresholds. Comparisons are expressed as tezepelumab versus dupilumab. Data are expressed as standardized mean differences (SMDs) and 95% credible intervals (CrIs). AQLQ, Asthma Quality of Life Questionnaire; Teze, tezepelumab; Dupi, dupilumab; PBEC, peripheral blood eosinophil count.

**Figure 7 cells-11-00819-f007:**
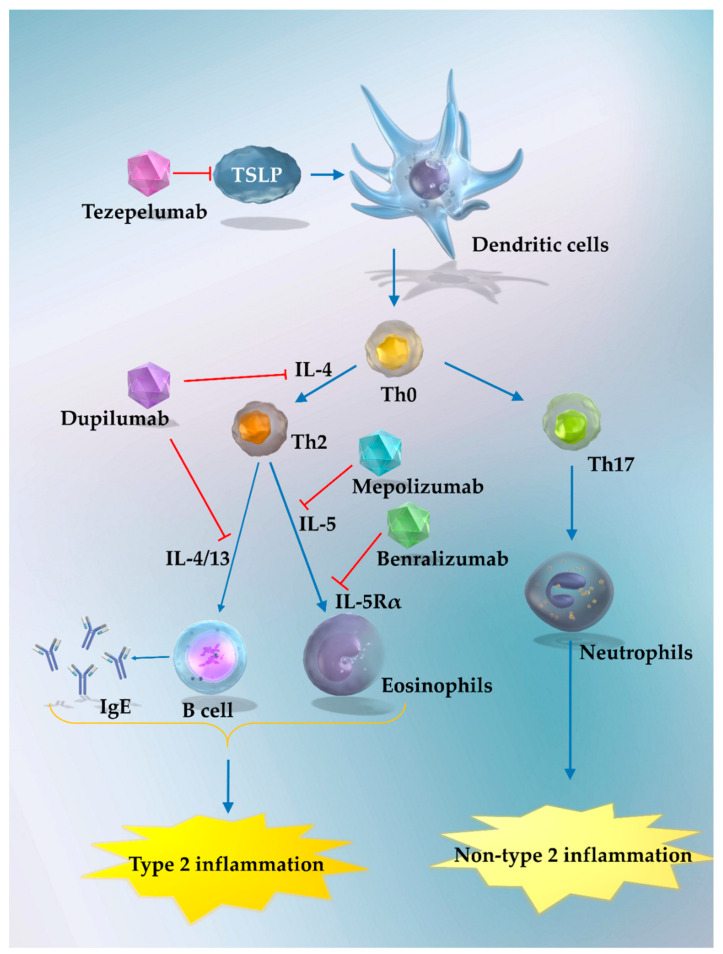
Overview of the **t**hymic stromal lymphopoietin (TSLP)-triggered immune response and pathogenesis of asthma. The cytokine TSLP is secreted by mucosal epithelial cells and its receptors are highly expressed on dendritic cells. TSLP induces a Th2-type immune response involving IL-4, IL-5, and IL-13 and is responsible for the pathogenesis of type 2 asthma. TSLPs are also involved in the pathogenesis of non-type 2 asthma by inducing the differentiation of Th17 cells by the activation of dendritic cells. Dupilumab targets IL-4 and IL-13, benralizumab targets IL-5Rα, and mepolizumab targets IL-5, and these drugs are mainly clinically effective in type 2 asthma. Tezepelumab targets TSLP and is expected to have clinical efficacy in both type 2 and non-type 2 asthma; IL, interleukin; IgE, Immunoglobulin E; IL-5Rα, interleukin-5 receptor alpha.

## Data Availability

The authors confirm that the data sets analyzed in this study are available from the corresponding author upon reasonable request.

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
