# Peer review of "Comparative Efficacy and Safety of Tezepelumab and Other Biologics in Patients with Inadequately Controlled Asthma According to Thresholds of Type 2 Inflammatory Biomarkers: A Systematic Review and Network Meta-Analysis"

_cells, 2022, doi:10.3390/cells11050819_

Round 1

Reviewer 1 Report

This aim is fully explained in the abstract and in the manuscript.  Moreover, the results are well defined and the English language is good.   In my opinion, this manuscript is suitable for the publication.

Author Response

Dear Reviewer:

We thank you for reviewing our manuscript titled “Comparison of the Efficacy and Safety of Tezepelumab and Other Biologics in Patients with Inadequately Controlled Asthma According to Thresholds of Type 2 Inflammatory Biomarkers: A Systematic Review and Network Meta-Analysis” (Manuscript ID: cells-1587359). We also thank you for your positive feedback on our manuscript.

Comment: This aim is fully explained in the abstract and in the manuscript.  Moreover, the results are well defined and the English language is good.   In my opinion, this manuscript is suitable for the publication.

Response: We thank you for reviewing our manuscript and for providing your positive feedback.

We are confident that our revised manuscript will be suitable for publication in Cells and look forward to receiving your editorial decision.

Thank you for your consideration.

Sincerely,

Koichi Ando

Department of Medicine, Division of Respiratory Medicine and Allergology, Showa University School of Medicine

1-5-8 Hatanodai, Shinagawa-ku, Tokyo, 142-8666, Japan

Tel: +81-3-3784-8532

Fax: +81-3-3784-8742

Reviewer 2 Report

The authors state in abstract and introduction that their analysis comprises 4 biologicals. Confusingly, they proceed to merely compare only 2 (Tezepelumab and Dupilomab) in the review. I suggest changing the abstract.

Yet, in the discussion they state that Tezepelumab ranked the best treatment amongst all biologicals. This is unwarranted especially since it is briefly mentioned that statistical significance was not achieved.

The rational for selection of Dupilumab seems week and without discussions of actual clinical data that support this strategy (only suggestions in previous reviews are referenced forwarded at the end of introduction)

Without critical discussion of prior art, the authors claim, in abstract and discussion, to have demonstrated a basis for deciding asthma treatment strategies. What is their evidence worth compared with the well-controlled individual studies in this field? and what is novel regarding treatment validations in previous studies?

The discussion of experimental cell data seems superficial and not specifically related to the primary end point in the clinical studies of annualized rates of asthma exacerbations (known to be generally caused/associated with respiratory viral infections).

Author Response

Dear Reviewer:

We thank you for reviewing our manuscript titled “Comparison of the Efficacy and Safety of Tezepelumab and Other Biologics in Patients with Inadequately Controlled Asthma According to Thresholds of Type 2 Inflammatory Biomarkers: A Systematic Review and Network Meta-Analysis” (Manuscript ID: cells-1587359). We also thank you for your valuable feedback on our manuscript. We have revised the manuscript according to the comments. The revised portions in the manuscript are highlighted in yellow. Our responses to the comments are provided below.

Comment 1: The authors state in abstract and introduction that their analysis comprises 4 biologicals. Confusingly, they proceed to merely compare only 2 (Tezepelumab and Dupilumab) in the review. I suggest changing the abstract.

Response: We thank you for the important comment, which we agree with. The abstract has been revised to reflect the contents of the main text more faithfully. We have included the key results of the comparison between tezepelumab and dupilumab and those of the ranking assessment among the five treatment groups namely tezepelumab, dupilumab, benralizumab, mepolizumab, and placebo. Furthermore, we have included the key results of the pairwise comparisons between each of the four biologics (tezepelumab, dupilumab, benralizumab, and mepolizumab) in the relevant parts of the abstract and results section. Moreover, in the last part of the introduction section, we have provided the rationale for this study, which focuses on the comparison between tezepelumab and dupilumab. We have also emphasized more clearly that the purpose of this study was to compare tezepelumab and other biologics (including benralizumab and mepolizumab, not just dupilumab) in general.

Comment 2: Yet, in the discussion they state that Tezepelumab ranked the best treatment amongst all biologicals. This is unwarranted especially since it is briefly mentioned that statistical significance was not achieved.

Response: We thank you for your valuable comment, which we agree with. As you pointed out, it is difficult to conclude the superiority or inferiority of each treatment based on the SUCRA value alone. In section 2 (Materials and Methods), we have added information that should be well-known and noted when the SUCRA values are interpreted (lines 163–171). In section 3 (Results), we describe only the SUCRA values that were objectively obtained by calculation and their relative rankings. In addition, in section 4 (Discussion; lines 479–485) and in the footnotes of Tables S5, S7, S9, S11, S13, and S15 that summarize the SUCRA results, we have added the caveat that the SUCRA values alone cannot verify the significance of the difference in efficacy of the drugs and that it was therefore not possible to make a final conclusion about these differences. Furthermore, in the discussion of limitations in section 4, we have added the following description: “Third, in this study, the SUCRA was used to rank the drugs. However, it should be noted that it is difficult to refer to the significance of drug efficacy based on the SUCRA values alone, and the ranking of SUCRA values is only an auxiliary reference finding. The comparison of efficacy and safety among drugs should be evaluated based on statistical significance.” (lines 533–537).

Comment 3: The rational for selection of Dupilumab seems week and without discussions of actual clinical data that support this strategy (only suggestions in previous reviews are referenced forwarded at the end of introduction)

Response: We thank you for your insightful comment, which we agree with. In this study, we compared five therapeutic intervention groups, namely, tezepelumab, dupilumab, benralizumab, mepolizumab, and placebo. Among them, we focused on the comparison of tezepelumab with dupilumab. The rationale for focusing on dupilumab has been added in more detail in the Introduction section (lines 97–106) and the discussion section (lines 442–448). Furthermore, we have more clearly mentioned in the last part of the introduction section that the purpose of this study was to compare tezepelumab with each biologic (including benralizumab and mepolizumab) (lines 107–109), not being limited only to compare tezepelumab with dupilumab. In addition to the comparison of tezepelumab and dupilumab, the comparison of tezepelumab and benralizumab, or comparison of tezepelumab and mepolizumab has also been described in the results section and the interpretation of the results has been presented in the discussion section (lines 451–455).

Comment 4: Without critical discussion of prior art, the authors claim, in abstract and discussion, to have demonstrated a basis for deciding asthma treatment strategies. What is their evidence worth compared with the well-controlled individual studies in this field? and what is novel regarding treatment validations in previous studies?

Response: We agree with your comment. We have revised the main text, added more description regarding the significance of our validation results, and highlighted the novelty of our validation compared with previous studies in the discussion section (lines 431–448).

Comment 5: The discussion of experimental cell data seems superficial and not specifically related to the primary end point in the clinical studies of annualized rates of asthma exacerbations (known to be generally caused/associated with respiratory viral infections).

Response: We thank you for this important comment. As you pointed out, respiratory viral infections are often known to trigger asthma exacerbations. Respiratory viral infections induce the production of TSLP from airway epithelial cells, leading to asthma exacerbations. Therefore, it is speculated that the inhibition of TSLP may inhibit the exacerbation of asthma associated with viral infection. A detailed explanation of the relationship between viral infection and the production of TSLP from the airway epithelium and its downstream signaling has been added to the introduction section (lines 66–69).

We are confident that our revised manuscript will be suitable for publication in Cells and look forward to receiving your editorial decision.

Thank you for your consideration.

Sincerely,

Koichi Ando

Department of Medicine, Division of Respiratory Medicine and Allergology, Showa University School of Medicine

1-5-8 Hatanodai, Shinagawa-ku, Tokyo, 142-8666, Japan

Tel: +81-3-3784-8532

Fax: +81-3-3784-8742

Reviewer 3 Report

That is a well-written manuscript comprehensively analyzing the clinical efficacy of tezepelumab compared to other biologics currently used in asthma therapy based on RCT results. The publication is interesting and important since tezepelumab seems to be the biological medication breaking through in non-T2 asthma.

I am a clinician; thus, I cannot properly validate the EBM methodology, although, in my opinion, it seems correct.

Comments:

1. The authors focused mainly on comparing dupilumab (model biologic for therapy of T2 asthma) in the main manuscript. However, they performed a study on all T2-related asthma biologics. Thus, the title of the manuscript is slightly confusing. I would suggest changing, also pointing to the other asthma biologics, e.g., Comparative Efficacy and Safety of Tezepelumab and Other Biologics ……  Such a topic may be more attractive; besides, it reflects the article's actual content.

2. The indirect comparisons between tezepelumab and benralizumab or tezepelumab and mepolizumab are provided mainly in Tables in the Supplementary file. Therefore, it is challenging to follow them. For example, Table S4 -  benralizumab is less effective in the overall population than tezepelumab (am I right?). The authors need to provide more details (at least conclusions) also in the main document regarding the comparisons of other medications, at least regarding AER, assessed as the primary efficacy endpoint. That would also be useful for secondary endpoints, such as AQLQ and ACQ scores. Furthermore, it is tough to understand the Supplementary Tables for someone unfamiliar with principles of systemic reviews of medical data. For example, Table S5, S7 – are there any significant differences among medications? If not, please provide an appropriate comment on that issue in the footnote or explain how to understand presented numbers. These results are also mentioned in the main Manuscript without comment on their statistical significance.  Please, correct. The same remark also regards other Tables in the Supplement. Please describe the main outcomes, their statistical and medical meanings in the Tables’ footnote. That will significantly improve the understanding of the work.

3. Figure 1: Please indicate in Fig. which JAK and STAT are likely activated. Different subtypes play different biological roles, e.g. STAT-1 is involved in Th1 and Th17 stimulation, whereas STAT-3 in T2 inflammatory response.  

4. Please explain all abbreviations in subtitles, eg.:  AER, PB, etc.

5. Figure 3 – please provide „Placebo” instead Plac – there is enough room for that

Author Response

Dear Reviewer:

We thank you for reviewing our manuscript titled “Comparison of the Efficacy and Safety of Tezepelumab and Other Biologics in Patients with Inadequately Controlled Asthma According to Thresholds of Type 2 Inflammatory Biomarkers: A Systematic Review and Network Meta-Analysis” (Manuscript ID: cells-1587359). We also thank you for your valuable feedback on our manuscript. We have revised the manuscript according to the comments. The revised portions in the manuscript are highlighted in yellow. Our responses to the comments are provided below.

Comment 1: The authors focused mainly on comparing dupilumab (model biologic for therapy of T2 asthma) in the main manuscript. However, they performed a study on all T2-related asthma biologics. Thus, the title of the manuscript is slightly confusing. I would suggest changing, also pointing to the other asthma biologics, e.g., Comparative Efficacy and Safety of Tezepelumab and Other Biologics ……  Such a topic may be more attractive; besides, it reflects the article's actual content.

Response: We thank you for your valuable suggestion, which we agree with. In other to reflect our study more faithfully, we have revised the title as follows:

"Comparison of the Efficacy and Safety of Tezepelumab and Other Biologics in Patients with Inadequately Controlled Asthma According to Thresholds of Type 2 Inflammatory Biomarkers: A Systematic Review and Network Meta-Analysis.”

Comment 2: The indirect comparisons between tezepelumab and benralizumab or tezepelumab and mepolizumab are provided mainly in Tables in the Supplementary file. Therefore, it is challenging to follow them. For example, Table S4 - benralizumab is less effective in the overall population than tezepelumab (am I right?). The authors need to provide more details (at least conclusions) also in the main document regarding the comparisons of other medications, at least regarding AER, assessed as the primary efficacy endpoint. That would also be useful for secondary endpoints, such as AQLQ and ACQ scores. Furthermore, it is tough to understand the Supplementary Tables for someone unfamiliar with principles of systemic reviews of medical data. For example, Table S5, S7 – are there any significant differences among medications? If not, please provide an appropriate comment on that issue in the footnote or explain how to understand presented numbers. These results are also mentioned in the main Manuscript without comment on their statistical significance. Please, correct. The same remark also regards other Tables in the Supplement. Please describe the main outcomes, their statistical and medical meanings in the Tables’ footnote. That will significantly improve the understanding of the work.

Response: We thank you for your valuable comment, which we agree with. Per your comment, in the results section, we have provided the results of comparison of tezepelumab and dupilumab for each outcome, comparison of tezepelumab and other biologics, and comparison of each pair of treatment groups other than tezepelumab including dupilumab, benralizumab, and mepolizumab (e.g., comparison of dupilumab and benralizumab or comparison of benralizumab and mepolizumab). In addition, in the footnotes of each supplementary table that summarizes the results, we explain how to interpret the results. For example, we have added the following explanation to the footnote of Table S4: “The effect size is expressed as Treatment A vs. Treatment B. If the 95% CrI value does not include 1, the difference in effect size between the drugs is considered to be significant. If the 95% CrI is localized in the range less than 1, Treatment A is more effective than Treatment B in terms of AER, whereas if the 95% CrI is located in the range greater than 1, Treatment A is less effective than Treatment B in terms of AER. For example, tezepelumab was found to be significantly more effective than benralizumab in the overall group and in the groups with PBEC ≥300 and ≥150, and more effective than dupilumab in the group with PBEC <150.” Furthermore, as the reviewers pointed out, SUCRA values alone could not verify the significance of the difference in efficacy of the drugs. In section 2 (Materials and Methods), we have added information that should be well-known and noted when the SUCRA values are interpreted (lines 163–171). In section 3 (Results), we describe only the SUCRA values that were objectively obtained by calculation and their relative rankings. In addition, in section 4 (Discussion; lines 479–485) and in the footnotes of Tables S5, S7, S9, S11, S13, and S15 that summarize the SUCRA results, we have added the caveat that the SUCRA values alone cannot verify the significance of the difference in efficacy of the drugs and that it was therefore not possible to make a final conclusion about these differences. Furthermore, in the discussion of limitations in section 4, we have added the following description: “Third, in this study, the SUCRA was used to rank the drugs. However, it should be noted that it is difficult to refer to the significance of drug efficacy based on the SUCRA values alone, and the ranking of SUCRA values is only an auxiliary reference finding. The comparison of efficacy and safety among drugs should be evaluated based on statistical significance.” (lines 533–537).

Comment 3: Figure 1: Please indicate in Fig. which JAK and STAT are likely activated. Different subtypes play different biological roles, e.g. STAT-1 is involved in Th1 and Th17 stimulation, whereas STAT-3 in T2 inflammatory response. 

Response: We agree with your comment. We thank you for pointing this out. We have modified Figure 1 and the relevant part in the introduction section (lines 49–50) to clearly present information about subtypes, and JAKs and STATs that are likely to be activated.

Comment 4: Please explain all abbreviations in subtitles, eg.:  AER, PB, etc.

Response: We thank you for pointing this out. We have ensured that all abbreviations used in the manuscript are properly defined in relevant parts.

Comment 5: Figure 3 – please provide „Placebo” instead Plac – there is enough room for that

Response: We agree with your comment. We have used "Placebo" instead of "Plac" in Figure 3.

We are confident that our revised manuscript will be suitable for publication in Cells and look forward to receiving your editorial decision.

Thank you for your consideration.

Sincerely,

Koichi Ando

Department of Medicine, Division of Respiratory Medicine and Allergology, Showa University School of Medicine

1-5-8 Hatanodai, Shinagawa-ku, Tokyo, 142-8666, Japan

Tel: +81-3-3784-8532

Fax: +81-3-3784-8742

Round 2

Reviewer 2 Report

The changes performed by the reviewers seem adequate now.

Reviewer 3 Report

Thank you for all improvements.

The Manuscript is much clarified now.